# Sense-overlapping lncRNA as a decoy of translational repressor protein for dimorphic gene expression

**Christelle Alexa Garcia Perez**[1], **Shungo Adachi**[2], **Quang Dang Nong**[1], **Nikko Adhitama**[1], **Tomoaki Matsuura**[1], **Toru Natsume**[2], **Tadashi Wada**[1], **Yasuhiko Kato**[1]*, **Hajime Watanabe**[1]*

1 Department of Biotechnology, Graduate School of Engineering, Osaka University, Suita, Japan, 2 Cellular and Molecular Biotechnology Research Institute (CMB), National Institute of Advanced Industrial Science and Technology (AIST), Tokyo, Japan

* watanabe@bio.eng.osaka-u.ac.jp (HW); kato_yasuhiko@bio.eng.osaka-u.ac.jp (YK)

**Data Availability Statement:** All relevant data are within the manuscript and its Supporting Information files.

## Abstract

Long noncoding RNAs (lncRNAs) are vastly transcribed and extensively studied but lncRNAs overlapping with the sense orientation of mRNA have been poorly studied. We analyzed the lncRNA *DAPALR* overlapping with the 5′ UTR of the *Doublesex1* (*Dsx1*), the male determining gene in *Daphnia magna*. By affinity purification, we identified an RNA binding protein, Shep as a *DAPALR* binding protein. Shep also binds to *Dsx1* 5′ UTR by recognizing the overlapping sequence and suppresses translation of the mRNA. *In vitro* and *in vivo* analyses indicated that *DAPALR* increased *Dsx1* translation efficiency by sequestration of Shep. This regulation was impaired when the Shep binding site in *DAPALR* was deleted. These results suggest that Shep suppresses the unintentional translation of *Dsx1* by setting a threshold; and when the sense lncRNA *DAPALR* is expressed, *DAPALR* cancels the suppression caused by Shep. This mechanism may be important to show dimorphic gene expressions such as sex determination and it may account for the binary expression in various developmental processes.

## Author summary

Long noncoding RNAs are vastly transcribed throughout the genome. Among them, RNAs overlapping the protein-coding RNA in sense orientation have been poorly studied because of the difficulty in differentiating their sequences from their overlapping coding RNAs although this class of RNAs has been reported to comprise the majority of the long noncoding RNAs. In the crustacean *Daphnia magna*, a long noncoding RNA, called *DAPALR*, is transcribed from the male determining gene, *Doublesex1*, and overlaps with the *Doublesex1* 5′ UTR. *DAPALR* activates *Doublesex1* but this regulatory mechanism remains unknown. We found the RNA binding protein Shep bound to the *Doublesex1* 5′ UTR. *In vitro* and *in vivo* experiments indicated that Shep suppresses translation of the mRNA and *DAPALR* increases *Doublesex1* translation efficiency by sequestration of Shep. Since male-specific expression of *Doublesex1* is also regulated at the transcriptional level,

**Funding:** This work was supported by 17H05610, 20H04853 to S.A., 20H04923, 19H05423, 18H04884 and 17H05602 to K.Y. and 18H04619, 17K19236 and 17H01880 to H.W. from Japan Science Promotion Society (JSPS), Japan (https://www.jsps.go.jp/english/). The funders had no role in study design, data collection and analysis, decision to publish, or preparation of the manuscript.

**Competing interests:** The authors have declared that no competing interests exist.

we propose that Shep cancels the unexpected expression of *Doublesex1* and maintains the feminized state for sexual dimorphism but *DAPALR* suppresses this repression by sequestration of Shep. We infer that this mechanism is not only for binary sex regulation but could function in the binary regulation of other genes in various biological processes.

## Introduction

Long noncoding RNAs (lncRNAs) are vastly transcribed in the genome and play a diverse role in the cell such as epigenetic regulation, transcription, and post-transcriptional regulation [1,2]. Based on the direction of the transcription of lncRNA, it can be categorized in its orientation as sense and antisense. While a growing knowledge about antisense lncRNA has been accumulated, knowledge about sense lncRNA is still limited. Especially, sense-overlapping lncRNAs that overlap protein-coding genes in the same sense strand remain poorly studied. This is despite the projection that sense-overlapping lncRNAs are actually the most abundant type of lncRNA based on the proportions of lncRNA classes in PacBio Iso-seq annotation [3].

Previously, we identified a sense-overlapping lncRNA called Doublesex1-alpha-promoter-associated-long noncoding-RNA (*DAPALR*) that can regulate *Doublesex1* (*Dsx1*) [4]. *Dsx1* is responsible for male determination in *Daphnia magna*. It consists of two isoforms, *Dsx1α* and *Dsx1ß* [5]. *DAPALR* is transcribed from upstream of the transcription start site of *Dsx1α* isoform and overlaps with its 5´ UTR [4]. Both isoforms of *Dsx1* and *DAPALR* are highly expressed in males and it has also been identified that *DAPALR* and its overlapping region with *Dsx1α* 5´ UTR can induce *Dsx1* expression in *trans* but its molecular mechanism remains unknown [4]. In this study, we identified the Shep as a *DAPALR* binding protein. Loss-of-function experiments and overexpression of *Shep* showed that Shep functions as a suppressor of *Dsx1*. *In vivo* and *in vitro* post-transcription assays showed that Shep binds to and represses the *Dsx1* mRNA and *DAPALR* sequesters Shep to activate the *Dsx1* translation.

## Results

### Identification of Shep as a sense lncRNA binding protein

As our previous study showed that the 205 bp of *DAPALR* fragment overlapping with *Dsx1α* 5´ UTR (Fig 1A) is the core region for the enhancement of *Dsx1* expression [4], we attempted to identify proteins that interact with the core region. We used the 205 bp overlapping sequence as bait for the RNA pulldown experiment. Through RNA pulldown using a FLAG-peptide tagged bait RNA incubated with *D. magna* lysate followed by mass spectrometry (Fig 1B) [6,7], we identified two candidate proteins: Alan shepard (Shep) and CUG binding protein 1 (CUGBP1). Among the pulled-down proteins, Shep and CUGBP1 resulted to high, significant p-values, which means that they have the highest probability for binding to the overlapping sequence of *DAPALR* as they did not associate with the negative bait samples like the *Dsx1ß* 5´ UTR (S1 Table). While both of the identified proteins are known to have RNA binding activities, we focused on Shep in this study because it has been reported to upregulate the expression of the target gene by suppressing the insulator activity [8] and Sup-26, the ortholog of Shep in *Caenorhabditis elegans*, regulates translation of the sex-determining gene *tra-2* [9].

We searched the *D. magna* genome database, *D. magna* Genome BLAST (http://arthropods.eugenes.org/EvidentialGene/daphnia/daphnia_magna/BLAST/), for the *Shep* ortholog and found a single *Shep* ortholog. It consists of 5 exons and codes for 458 amino acids of a polypeptide including the two RNA Recognition Motifs (RRMs) (Fig 1C). The

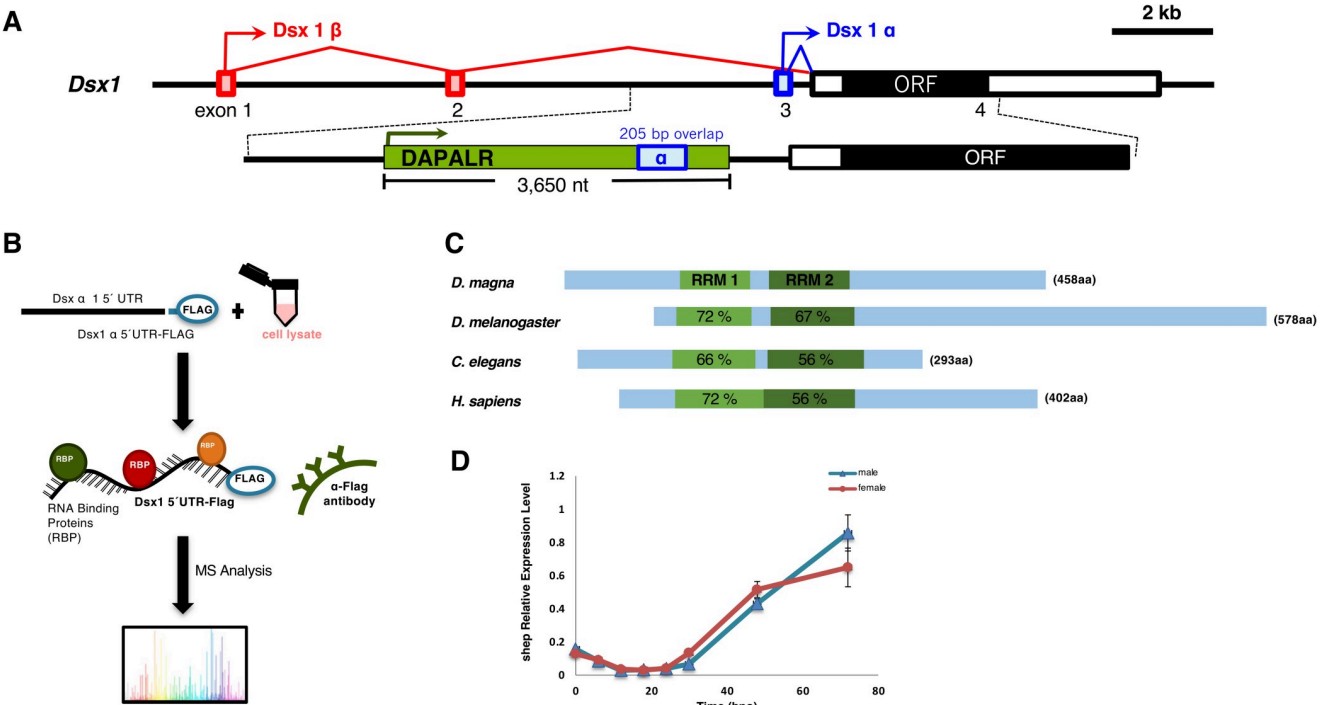

**Fig 1. Identification of Alan Shepard (Shep) as an RNA binding protein of *DAPALR*.** **(A)** Genomic organization of *Dsx1* gene in *Daphnia magna*. Exons are indicated by boxes; red: *Dsx1 β*, blue: *Dsx1 α*. The ORF is indicated by a black box. The position and orientation of *DAPALR* are indicated by the green box and arrow. **(B)** Outline of the RNA pulldown assay for the identification of the RNA binding proteins. **(C)** Domain structure of *Shep* in *D. magna* and its similarity with its orthologs in other species. The blue boxes represent the Shep ORF. The two RRMs (RNA Recognition Motifs) domains are shown in green boxes within the ORF. **(D)** Temporal expression profile of *Shep* in embryonic developmental stages. Results are shown as relative expression normalized with the ribosomal protein L32. Error bars indicate the standard error of the mean (n = 3), not significant in all points (Student's T-test between male and female).

multiple sequence alignment with other *Shep* orthologs demonstrated that the RRMs are highly conserved among species and throughout evolution (Figs 1B, S1 and S2).

As we identified the Shep as a *DAPALR* binding protein and *DAPALR* shows sexually dimorphic expression, we examined if *Shep* also shows sexual dimorphism. While *Dsx1* and *DAPALR* both have male-specific expression, *Shep* was expressed both in male and female embryos and did not exhibit sexual dimorphism (Fig 1D). The expression of *Shep* started to increase after 30 hours post-ovulation (hpo), mirroring the expression pattern of *Dsx1* in males [5].

## Knockdown of *Shep* mRNA enhances the *Dsx1* expression

To elucidate the functions of *Shep*, loss-of-function analyses were performed using the *Dsx1* reporter strain [10]. In this strain, the *mCherry* gene was inserted at the translation initiation codon of the endogenous *Dsx1* gene in one allele, in addition to the ubiquitous expression of the H2B-GFP. We injected the *Shep*-targeting siRNA into the eggs obtained from the *Dsx1* reporter strain and found that *Shep* knockdown resulted in a 5-fold and 3-fold increase of mCherry fluorescence in female and male embryos, respectively (Fig 2A and 2B). The enhanced mCherry expression pattern by *Shep* RNAi (S3A Fig) was similar to that of *DAPALR* overexpression in female [4] and male embryos (S3B Fig). In male embryos, the enhanced mCherry signals could be observed not only in its male-specific organs such as the first antennae and its thoracic appendages but ubiquitously in its whole body (Fig 2A). While in female,

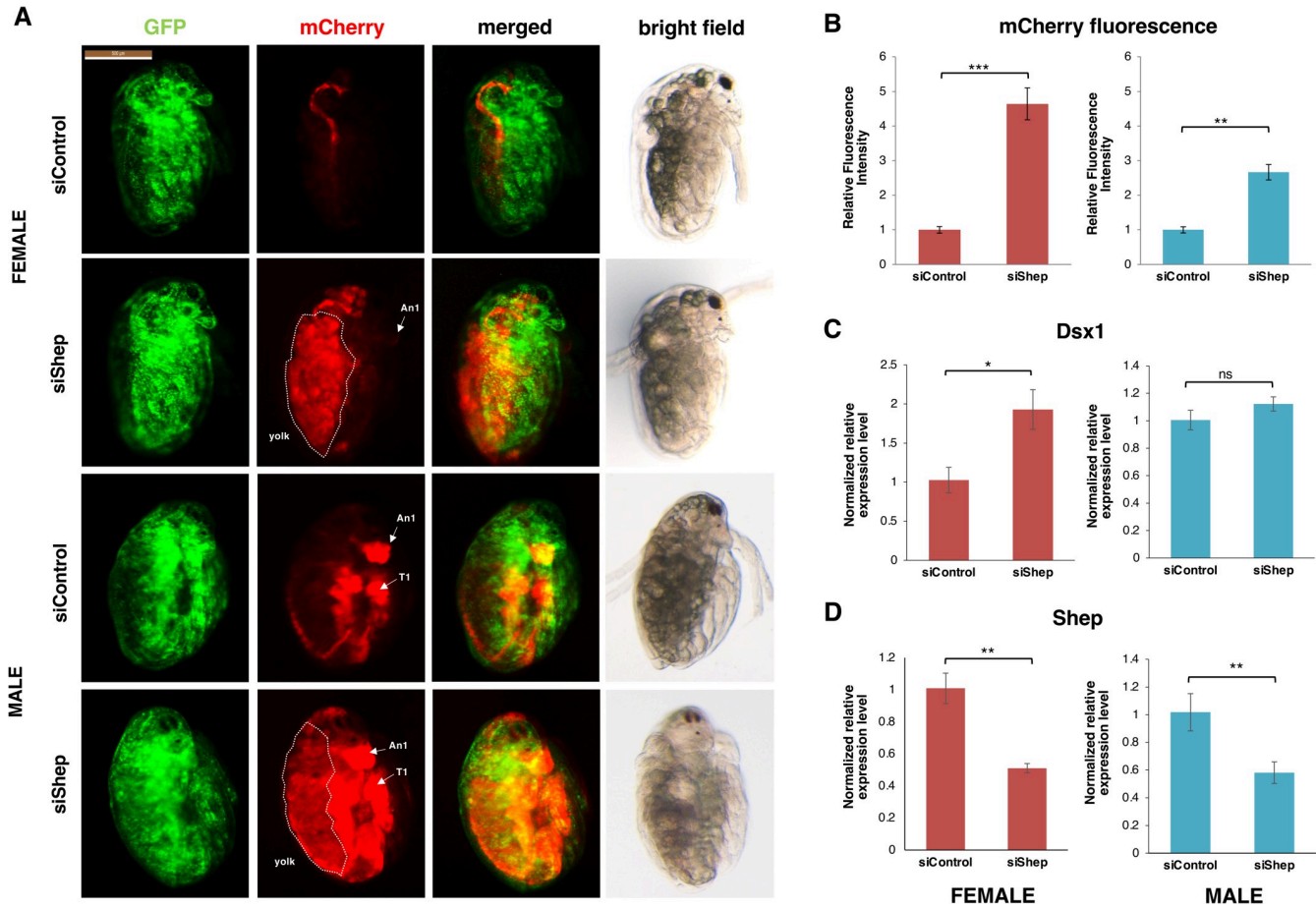

**Fig 2. Shep loss of function analysis. (A)** Lateral view of female and male embryos of *Dsx1* reporter strain injected with control siRNA and *Shep* siRNA and observed at 48 h after injection. mCherry fluorescence allowed visualization of *Dsx1* expression while GFP fluorescence in the nucleus enabled observation of body structures. The merged images of mCherry and GFP and the bright field images were used to understand the localization pattern of mCherry expression. An1: first antennae, T1: first thoracic leg, dotted lines: yolk area. **(B)** Relative mCherry fluorescence intensity calculated between *Shep* siRNA- and control siRNA-injected female (red) and male (blue) embryos. Error bars indicate the standard error of the mean (n = 5). **(C)** Gene expression profile of *Dsx1* in control siRNA- and *Shep* siRNA-injected female (red) and male (blue) embryos. **(D)** Gene expression profile of *Shep* in control siRNA- and *Shep* siRNA-injected female (red) and male (blue) embryos. RT-qPCR results are shown as expression levels normalized with housekeeping genes *L32*, *L8* and *Cyclophilin* and relatively compared to the control. Error bars indicate the standard error of the mean (n = 3). *p<0.05, **p<0.01, ***p<0.001, ns: not significant (Student's T-test).

high expression of mCherry signals was observed especially in the yolk region (Fig 2A). However, the mCherry fluorescence recapitulating the *Dsx1* expression was not enhanced in the male specific organs such as first antennae, showing that sex reversal did not occur.

In contrast to the prominent enhancement of the mCherry signals, no significant increase of *Dsx1* transcript was observed in males (Fig 2C) and a two-fold increase of *Dsx1* transcript level was observed in females (Fig 2C). Reduction of the *Shep* mRNA by RNAi (Fig 2D) suggested that the the mCherry enhancement was due to the reduction of Shep protein. The finding that the *Dsx1* transcription level does not reflect the enhancement of the mCherry expression suggests the possibility that Shep suppresses the translation of *Dsx1*.

## *Shep* mutant enhances the *Dsx1* expression

Next, we tried to introduce a mutation in the *Shep* gene using the CRISPR/Cas system. We used two types of gRNAs targeting each RRM, injected those gRNAs with Cas9 protein into

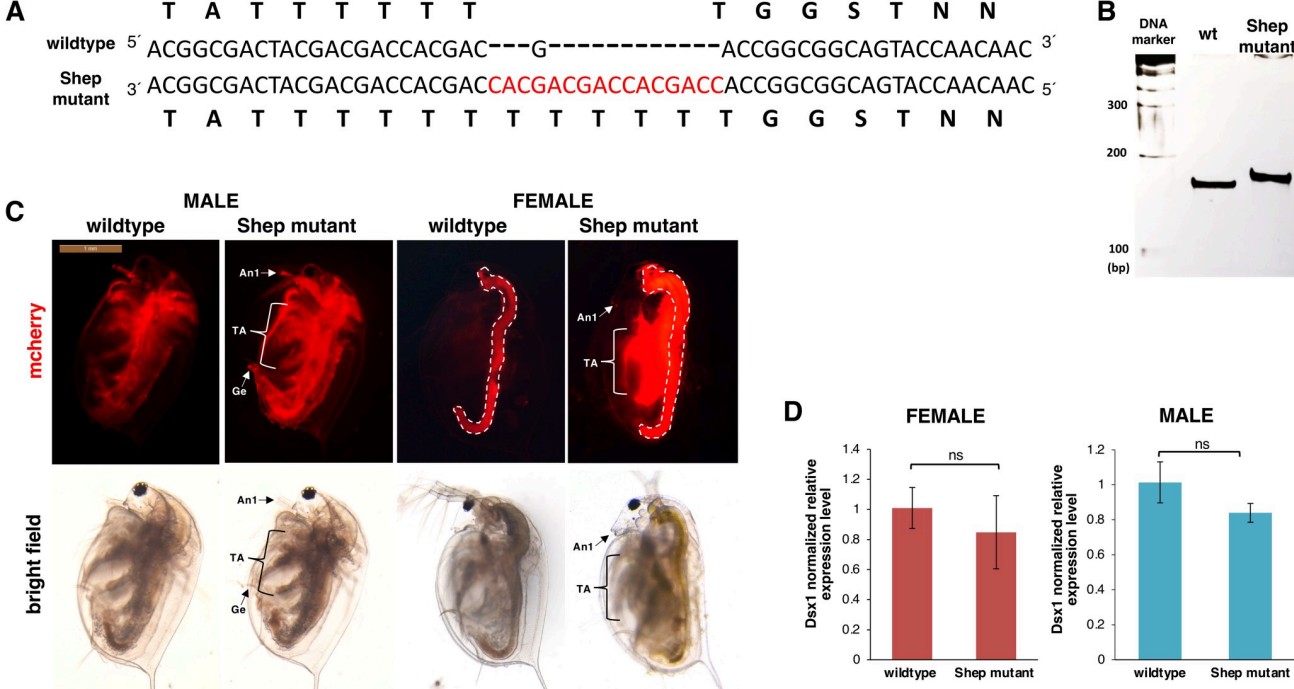

**Fig 3. Generated *Shep* mutant line. (A)** Nucleotide and amino acid sequence comparison between wildtype and *Shep* mutant. **(B)** PAGE analysis of PCR products by genomic PCR to amplify the Cas9/gRNA targeting region in the *Shep* coding sequence in the *Shep* mutant line. **(C)** Lateral view of the 2w *Shep* male and female mutant lines showing increase in mCherry fluorescence. The red signal from the guts (dotted lines) represents the autofluorescence of *Chlorella*, the main food used in daphniid cultivation. An1: first antennae, TA: thoracic appendages, Ge: genital. **(D)** Gene expression profile of *Dsx1* in 1w female and male wildtype control and *Shep-*. RT-qPCR results are shown as expression levels normalized with housekeeping genes *L32*, *L8* and *Cyclophilin* and relatively compared to the wildtype control. Error bars indicate the standard error of the mean (n = 3). ns: not significant (Student's T-test).

eggs from the *Dsx1* reporter strain, and obtained a line that has 15 nt insertion just before the RRM1 domain-coding sequence (Fig 3A, 3B and S2 Table). This line could be maintained and they developed normally into adults, producing offspring.

There were no noticeable differences between the mutant and wildtype at embryonic stages (S4 Fig). We also observed the mCherry expression of mutant daphniids at the adult stage when sexually dimorphic traits are more evident [10]. In the *Shep* mutant, both males and females showed significantly higher mCherry fluorescence than the wildtype (Fig 3C). Female daphniids of the *Dsx1* reporter strain do not usually have mCherry fluorescence [10], but the *Shep* mutant displayed mCherry signals in its whole body especially the appendages. High expression of mCherry signal could also be observed in the first antennae that is one of the major male-specific traits. However, the first antennae did not develop elongated like in males, signifying that sex reversal and male differentiation did not occur. On the other hand, the male mutant showed increased mCherry signals not only in male-specific regions such as the first antennae and genital but also in other regions. These suggest that Shep may suppress *Dsx1* in both male and female. In contrast to the drastic difference of the mCherry expression between the *Shep* mutant line and wildtype, we could not find a significant difference in *Dsx1* mRNA expression levels between the *Shep* mutant and wildtype (Fig 3D) in either male or female. This finding also supports the possibility that Shep controls *Dsx1* expression at post-transcriptional levels.

Using the CRISPR/Cas system, we first aimed to produce *Shep* mutant lines that have deletion mutations in the RRM domain. However, the embryos with indel mutations in both of

the two RRM domains could not hatch and they exhibited delayed or deformed phenotypes (S5 Fig: delayed development, deformed embryos, and unhatched eggs), suggesting that Shep is also essential for development and morphogenesis.

## Shep overexpression suppresses *Dsx1* expression

As our findings suggested that the diminished function of Shep increased the mCherry expression at the post-transcriptional level, we further investigated if *Shep* overexpression can suppress the *Dsx1* expression. We injected *in vitro* transcribed *Shep* mRNA into male eggs obtained from the *Dsx1* reporter line. As a result, we found that mCherry fluorescence was reduced in the *Shep* mRNA injected embryos (Fig 4A and 4B). *Shep* mRNA level was confirmed to increase after injection (Fig 4C) but the transcript level of *Dsx1* did not show any significant difference from the control (Fig 4D). These results also suggest that Shep does not affect *Dsx1* transcription or change the mRNA stability; rather its effect is at the translational level.

## TGE element is responsible for the post-transcriptional regulation or *DAPALR* function

As the *C. elegans* ortholog of Shep, Sup-26 has been reported to bind to the target sequence named <u>t</u>ra-2 and <u>GLI</u> element (TGE) to regulate the *Tra2* gene translation [9], we searched a similar sequence to the TGE in *Dsx1α* 5´ UTR and *DAPALR*. In the overlapping region of *Dsx1α* 5´ UTR and *DAPALR*, a highly conserved sequence with TGE was found (Fig 5A). To prove that the TGE-like motif is essential for the Shep function in *Daphnia*, either 40 nt of RNA including the potential TGE, or the 30 nt RNA that lacks the potential TGE was overexpressed in female embryos of the *Dsx1* reporter strain. When the RNA containing the TGE-like motif was expressed, the mCherry expression could be observed. The enhancement of the mCherry expression was the same result as the *DAPALR* overexpression [4]. In contrast, the deleted TGE did not have any effect on the reporter mCherry expression (Fig 5B), which was the same result as the injection of unrelated RNA. These results suggest the possibility that the TGE-like motif has a potential role in the function of Shep and *DAPALR* in *Dsx1* regulation.

## Shep binding site (TGE) is a target sequence of translational regulation

To test our hypothesis that the potential Shep binding site located in the 5´ UTR is the target of translational regulation of *DAPALR* and that the Shep functions as a translational suppressor, we examined translational efficiency in the presence or absence of Shep binding site in the mRNA. The GFP reporter mRNA harboring the *Dsx1α* 5´ UTR and the same mRNA only lacking the potential Shep binding site were prepared. These reporter mRNAs were individually injected into female wild-type eggs. Results showed that mRNA lacking TGE-like motif showed much higher expression of the GFP than wildtype mRNA (Fig 5C and 5D), suggesting that endogenous Shep may suppress the translation by binding to the *Dsx1α* 5´ UTR. Significant reduction of the GFP fluorescence was observed when *Shep* mRNA was co-injected with the intact Dsx1α 5´ UTR::GFP reporter mRNA (Fig 5C and 5D), indicating that Shep functions at the post-transcriptional level by suppressing translation.

## Shep binds the TGE for translational repression of *Dsx1*

To confirm that Shep binds to and regulates *Dsx1* translation through the TGE-like motif, we performed the suppression experiment *in vitro*. The luciferase gene was fused to two *Dsx1α* 5´ UTRs, one with an intact TGE-like motif and the other with the deleted TGE

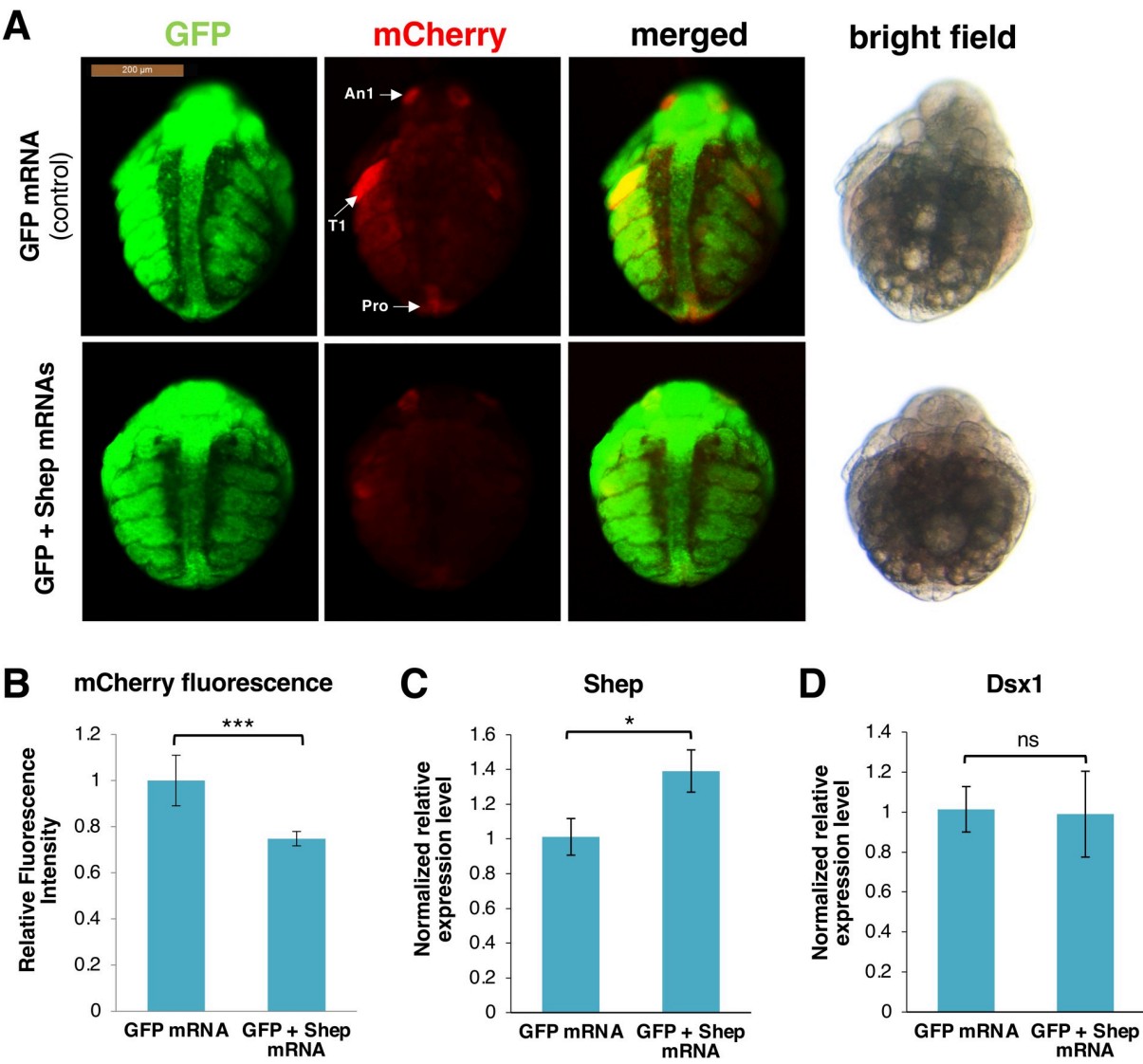

**Fig 4. Overexpression of Shep.** **(A)** Ventral view of male embryos of *Dsx1* reporter strain injected with GFP mRNA as control and GFP plus *Shep* mRNA observed at 30 h after injection. mCherry fluorescence allowed visualization of *Dsx1* expression while GFP fluorescence in the nucleus enabled observation of body structures. The merged images of mCherry and GFP and the bright field images were used to understand the localization pattern of mCherry expression. An1: first antennae, T1: first thoracic legs, Ge: genital. **(B)** Relative mCherry fluorescence intensity calculated between *GFP* mRNA- and GFP plus *Shep* mRNA-injected male embryos. Error bars indicate the standard error of the mean (n = 5). **(C)** Gene expression profile of *Shep* in *GFP* mRNA- and *GFP* plus *Shep* mRNA-injected male embryos. **(D)** Gene expression profile of *Dsx1* in *GFP* mRNA- and *GFP* plus *Shep* mRNA-injected male embryos. RT-qPCR results are shown as expression levels normalized with housekeeping genes *L32*, *L8* and *Cyclophilin* and relatively compared to the control. Error bars indicate the standard error of the mean (n = 3). $^*$p<0.05, $^{***}$p<0.001, ns: not significant (Student's T-test).

sequence. The mRNAs were synthesized *in vitro* and were translated with or without the *Shep* mRNA. The luciferase activity of the *Dsx1* reporter mRNA which harbors the Shep binding site was significantly reduced by the addition of *Shep* mRNA (Fig 6A). While the translation of the reporter mRNA without the Shep binding site remained unaffected by the presence of Shep, indicating that Shep suppresses the translation of the reporter mRNA through the TGE-like motif.

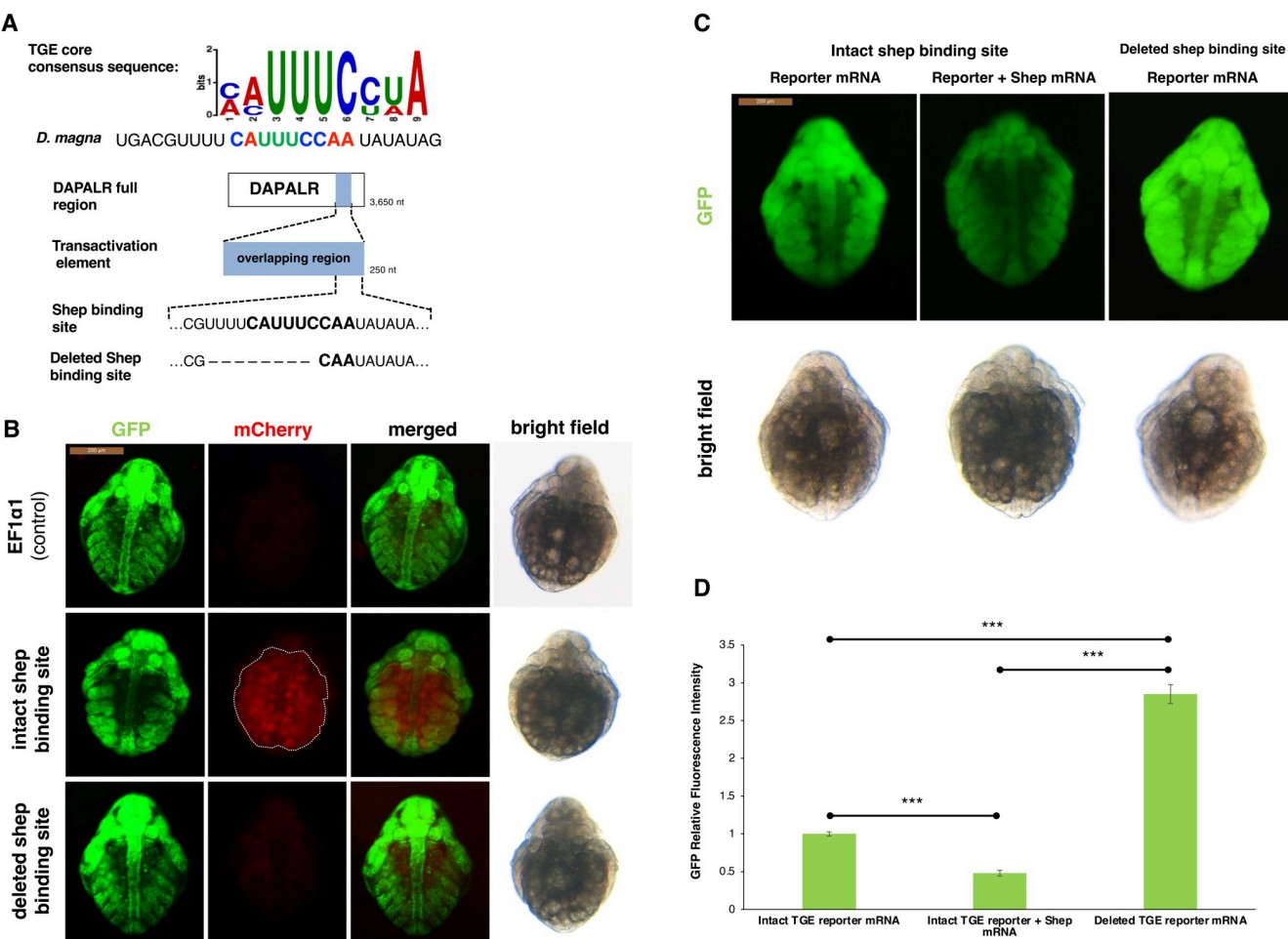

**Fig 5. *Dsx1* post-transcription regulation by *DAPALR* and *Shep in vivo*. (A)** Shep binding site consensus sequence and its similarity with TGE core consensus sequence. Position of the potential binding site was also shown located at the 3′ end of the transactivation element of *DAPALR*. Sequence of the mutated Shep binding site used for the experiment was also shown. **(B)** Ventral view of female embryos of *Dsx1* reporter strain injected with control plasmid, plasmid expressing intact TGE and plasmid expressing deleted TGE. mCherry fluorescence allowed visualization of *Dsx1* expression while GFP fluorescence in the nucleus enabled observation of body structures. The merged images of mCherry and GFP and the bright field images were used to understand the localization pattern of mCherry expression. dotted lines: yolk area. **(C)** Ventral view of female embryos of wildtype strain injected with *Dsx1* 5′ UTR-GFP reporter mRNA and reporter mRNA plus *Shep* mRNA and *Dsx1* 5′ UTR without TGE-GFP reporter mRNA observed at 30 h after injection. GFP fluorescence signals showed efficiency of translation. The bright field images were used to understand the localization pattern of GFP expression. **(D)** Relative GFP fluorescence intensity calculated among three treatments. Error bars indicate the standard error of the mean, n = 5. The end points of the line above the bars show which samples were compared statistically. ***p<0.001(Student's T-test).

To further confirm the direct interaction of Shep and its proposed binding site, we conducted a pulldown experiment using a FLAG-tagged Shep. We first confirmed that the FLAG-tagged Shep functioned the same as the wildtype Shep in suppressing the *Dsx1* reporter mRNA in the presence of the TGE while the unrelated FLAG-tagged protein (Flag-EcR) as the negative control showed no effect on the translation of *Dsx1* with or without the TGE (Fig 6A). The luciferase reporter mRNA harboring the intact TGE motif and another RNA without the motif were separately incubated to interact with the *in vitro* translated FLAG-tagged Shep and other controls, the Shep and the unrelated FLAG-tagged EcR. After pulldown using the M2 Anti-FLAG Affinity Gel, only the RNA with the intact TGE in the Shep-FLAG treatment showed highly significant enrichment (Fig 6B). The RNA without the TGE failed to bind with the Shep-FLAG, proving the exclusive binding of Shep to the RNA harboring the TGE. These

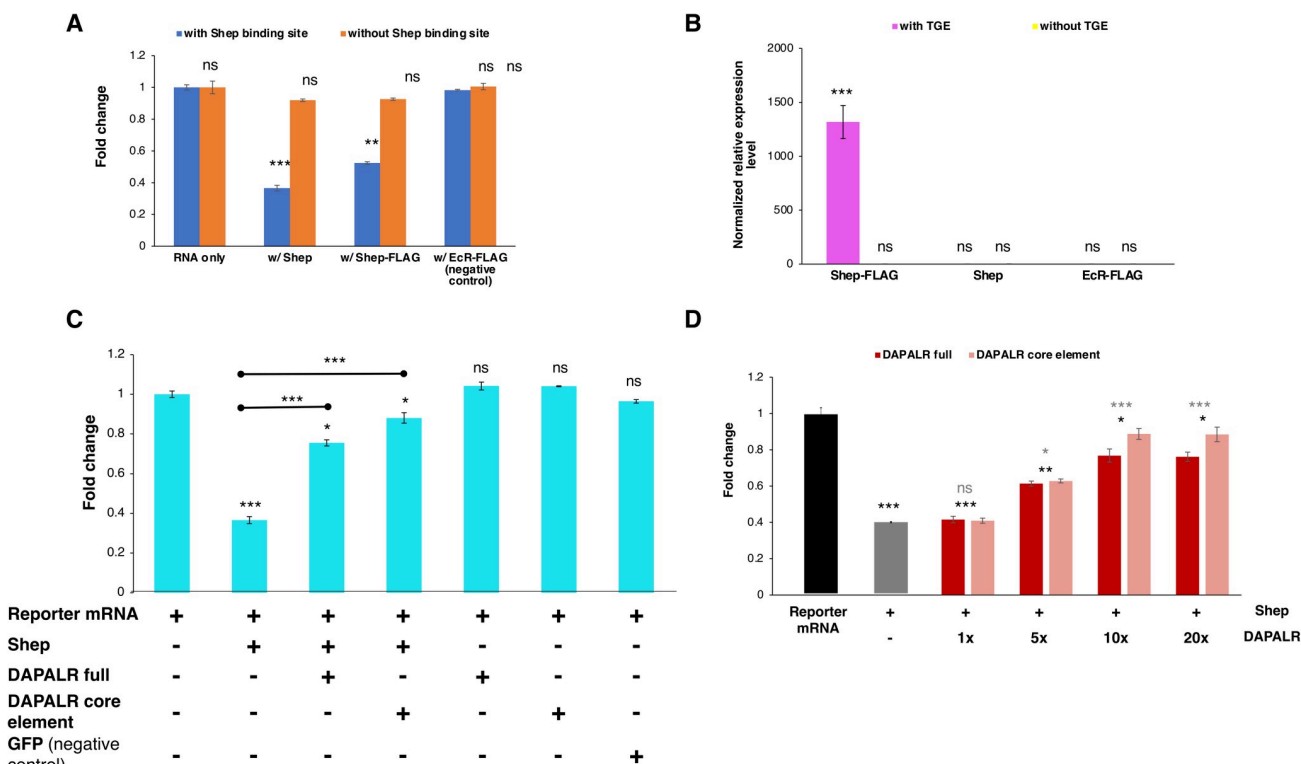

**Fig 6. *Dsx1* post-transcription regulation by *DAPALR* and Shep *in vitro*. (A)** Relative luciferase activity after *in vitro* translation assay of *Dsx1* 5′ UTR-Luc reporter mRNA with intact TGE and *Dsx1* 5′ UTR-Luc reporter mRNA without the TGE upon addition of *Shep* mRNA, *Shep-FLAG* mRNA and *EcR-FLAG* mRNA (negative control). Samples were compared against the expression of the *Dsx1* 5′ UTR-Luc reporter mRNA with intact Shep binding site without the addition of any other mRNAs. **(B)** Enrichment of the RNAs with and without the TGE after FLAG pulldown assay. Samples are compared against the negative control, RNA without the Shep binding site pulled using EcR-FLAG. **(C)** Relative luciferase activity after *in vitro* translation assay of *Dsx1* 5′ UTR-Luc reporter mRNA with intact TGE upon addition of *Shep* mRNA, *DAPALR* full RNA, *DAPALR* core element and *GFP* mRNA (negative control). Samples were compared against the expression of the *Dsx1* 5′ UTR-Luc reporter mRNA without the addition of any other mRNAs. The endpoints of the line above the bars show which sample were additionally compared statistically. **(D)** Relative luciferase activity after *in vitro* translation assay of *Dsx1* 5′ UTR-Luc reporter mRNA with Shep and different concentrations of full region of *DAPALR* and its core element. Error bars indicate the standard error of the mean, n = 3. Black asterisks show significant statistics compared with the expression of the *Dsx1* 5′ UTR-Luc reporter mRNA. Gray asterisks show significant statistics compared with the Reporter mRNA with Shep. Error bars indicate the standard error of the mean, n = 3. *p<0.05, **p<0.01, ***p<0.001, ns: not significant (Student's T-test).

results support that the TGE-like motif is indeed the Shep binding site and it is through this binding site that Shep regulates the *Dsx1* translation.

## *DAPALR* regulates translation efficiency

To exhibit the *DAPALR*-Shep regulation at the translational level, we performed the suppression experiment *in vitro* with the addition of the full region of *DAPALR* and its partial region harboring the core element that has the Shep binding site. Consistent with the *in vivo* experiment results, the addition of either the *DAPALR* or its core element to the *Dsx1* reporter mRNA with the Shep binding site, enhanced the luciferase translation activity even in the presence of Shep (Fig 6C). Different concentrations of the full and core element of *DAPALR* were also tested and results showed that the effect of the two *DAPALR* versions were not significantly different from one another and their rescue efficiencies were both dose-dependent (Fig 6D). The concentration of *DAPALR* needed to be at least 5 times higher than the reporter mRNA and Shep to be able to observe its decoy activity. These results showed the role of *DAPALR* in canceling the suppression of Shep to *Dsx1* translation.

## Discussion

Amidst the increasing knowledge of lncRNAs, the function of sense overlapping lncRNA is still lacking. Here we investigated function of the Shep as a key player to harness the lncRNA and the gene expression of *Dsx1*. In females where *Dsx1* is transcriptionally silenced, *Shep* loss-of-function increased the Dsx1 expression but it was still not as high in the manipulated females compared to that in males. Therefore, the *Shep* loss-of-function did not lead to sex reversal from female to male. In males, the *Dsx1* expression was enhanced by the *Shep* loss-of-function throughout the body. Importantly, *DAPALR* overexpression led to similar change of the *Dsx1* expression pattern both in females [4] and males (S3B Fig). To understand functional relationship on *Dsx1* expression between Shep and *DAPALR*, we also performed *in vitro* experiments. The FLAG pulldown experiment showed the exclusive binding of Shep to the TGE and suppression experiment showed that Shep inhibits the translation of *Dsx1* in presence of the TGE. Moreover, addition of *DAPALR* relieved the suppression caused by Shep and activated *Dsx1* translation.

Based on the results, we propose the noise canceling mechanism as a function of the sense overlapping lncRNA and Shep. In females, *Dsx1* transcription is repressed for avoiding masculinization. However, due to stochasticity in gene expression [11,12], there would be the noise in gene expression. In a previous study, we proved that improper expression of *Dsx1* changes the expression profile of its downstream genes resulting in intersex [13]; which suggested that the stochastic transcription of *Dsx1* causing population heterogeneity, should be avoided. In the presence of the Shep, the *Dsx1* mRNA from the transcriptional noise cannot be translated immediately because of the binding of the Shep at the TGE-like motif. When the sense overlapping lncRNA *DAPALR* is expressed, the Shep is sequestered from the mRNA by the *DAPALR* and the *Dsx1* translation is unlocked. This mechanism may function to avoid the unexpected expression of the *Dsx1* to accomplish sexual dimorphic expression.

In our decoy model, a quantitative relationship between Shep, *DAPALR*, and *Dsx1* mRNA needs to be considered. Although quantitative estimation of Shep in the *DAPALR*- and *Dsx1*-expressing cells is difficult, we assume that the quantity of Shep in the cell may define a threshold to cancel the effect of noisy transcription and the stochastic transcript below the threshold may not be translated because of the presence of Shep. The copy number of the *DAPALR* is one-tenth of the *Dsx1* mRNA [4], which may be a sufficient quantity to unlock the Shep suppression. The less abundance of *DAPALR* may be related to the more localized expression in comparison to the *Dsx1* because the extracted total RNA for qPCR was from the whole embryos. In this scenario, *DAPALR* may be expressed in cells only at the early stage of male differentiation and decrease the threshold of *Dsx1* expression by sequestering the Shep protein. Then, the translated Dsx1 protein may activate its own promoter by a positive feedback loop to maintain Dsx1 expression. And since *Shep* seems to be ubiquitously expressed in different tissues in the whole body, this RBP may be able to silence Dsx1 expression in non-sexually dimorphic tissues that do not express *DAPALR*. This hypothesis is consistent when *DAPALR* was overexpressed ubiquitously or Shep expression was silenced by siRNA and the *Dsx1* reporter mCherry expression was observed throughout the body even in the non-sexually dimorphic tissues. Further studies to prove this hypothesis could be investigated in the future.

Interestingly, both *Dsx1* and *Tra-2* are key regulators of sex determination [14] and they should be strictly regulated to avoid sexual ambiguity. Although there is no knowledge about the sense lncRNA at *Tra-2* locus, a similar mechanism may function in *Drosophila*. The Shep is also known to function in neurogenesis [15] and the Shep functions at a translational level [16]. It may also be possible that the Shep suppresses the translation under the control of unknown lncRNA and the genes whose noisy expression is harmful to the cell may have such

kind of noise-canceling system. The tight regulation of *Dsx1* through Shep-dependent suppression and by lncRNA exhibits one mechanism of how nature keeps intersex and sexual ambiguity rare.

Shep has been reported to have many functions such as antagonizing chromatin insulator activity, transcriptional and post-transcriptional control, especially in neurogenesis [8,16,17]. Localization of Shep in the cytoplasm [9] also supports the possibility that Shep functions at a post-transcriptional level.

In this study, we focused on the Shep and found that the Shep functions as a noise canceler and the *DAPALR* unlocks it. For further understanding of the *DAPALR*, the other RNA binding protein, CGUBP1, should be considered to understand the robust sex-determination system in *D. magna*.

A similar mechanism is known in the lncRNA named *linc-MD1*, in which miRNA is sequestered from the mRNA by the lncRNA [18], and a competing endogenous RNA hypothesis has been proposed [19]. Our finding suggests that the RBP such as the Shep can be a target of competing endogenous RNA in a broad meaning.

In a previous study enumerating a list of post-transcription regulators that Shep binds to, 5 out 77 Shep targets are noncoding RNAs [16]. While none of which mechanisms have been studied extensively and that the three-way network of translation regulation involving an mRNA, noncoding RNA, and RBP may be the first involving Shep; this regulation may occur more commonly. It is predicted that the sense-overlapping lncRNAs comprise the majority of the lncRNA present [3]. And as Shep is expressed not only in neurons but other tissues [15,16], the unique role of *DAPALR*-Shep-*Dsx1* may not only be for binary sex ultrasensitivity but also for binary regulation of other genes in various biological processes.

## Materials and methods

### *Daphnia magna* strains and transgenic lines culture

All of the wild-type (WT) and transgenic *Daphnia magna* lines share the same genetic background (NIES strain) and were cultured in AdaM medium [20] as previously described [5]. The transgenic line mostly used was the *Dsx1*-reporter strain that has the mCherry gene introduced upstream of the *Dsx1* coding sequence [10]. This line also has GFP fused to histone H2B gene under the control of the elongation factor 1α1 promoter/enhancer. Another transgenic line was established from crossing the *Dsx1*-reporter line to wildtype and finally choosing the progeny that does not have the H2B-EGFP gene. Male daphniids were obtained by exposing 2–3 weeks old female to 1 μg/L of the synthetic JH analog Fenoxycarb (Wako Pure Chemical, Osaka, Japan) [21].

### Preparation of bait RNAs and RNA pulldown assay

Preparation of Flag peptide conjugated bait RNAs were carried out as described previously [6]. Briefly, the T7-tagged cDNA template was amplified by the polymerase chain reaction (PCR), transcribed in vitro using the MEGAscript T7 kit (Invitrogen, Carlsbad CA, USA) and purified with an RNeasy Mini Kit (Qiagen). The 3′ end of purified cRNA was dialdehyded with 0.1 M $NaIO_4$, precipitated with 2% $LiClO_4$ in acetone and then washed with acetone. The pellet was dissolved in 0.1 M sodium acetate, pH 5.2 and then mixed with 30 mM hydrazide–Flag peptide. The resulting imine-moiety of the cRNA was reduced by adding 1 M $NaCNBH3$. The Flag -tagged-RNA was purified with an RNeasy Mini Kit (Qiagen).

For the pulldown assay, 1- or 2-day-old female larvae were lysed with lysis buffer [10 mM 4-(2-hydroxyethyl)-1-piperazineethanesulfonic acid (HEPES) (pH 7.5), 150 mM NaCl, 50 mM NaF, 1 mM $Na3VO4$, 5 μg/ml leupeptin, 5 μg ml aprotinin, 3 μg/ml pepstatin A, 1 mM

phenylmethylsulfonyl fluoride (PMSF), 1 mg/ml digitonin] using pre-chilled Dounce homogenizer (type A pestle) and cleared by centrifugation. One mg of cleared lysate was incubated with five pmol of Flag-tagged bait RNA, anti-FLAG antibody (Sigma) and protein G conjugated magnetic beads (Thermo) rotate for 1h at 4˚C. The magnetic beads were then washed three times with wash buffer [10 mM HEPES (pH 7.5), 150 mM NaCl, 0.1% Triton X-100] and co-immunoprecipitated RNA and proteins were eluted with Flag elution buffer [0.5 mg/ml Flag peptide, 10 mM HEPES (pH 7.5), 150 mM NaCl, 0.05% Triton X-100]. The bait RNA-associated proteins were then precipitated with TCA. Precipitated protein was re-dissolved in guanidine hydrochloride and reduced with TCEP, alkylated with iodoacetamide, followed by digestion with lysyl endopeptidase and trypsin. The digested peptide mixture was applied to a Mightysil-PR-18 (Kanto Chemical) frit-less column (45 3 0.150 mm ID) and separated using a 0–40% gradient of acetonitrile containing 0.1% formic acid for 80 min at a flow rate of 100 nL/min. Eluted peptides were sprayed directly into a mass spectrometer (QSTAR Elite, Sciex). The mass spectrometry and tandem mass spectrometry spectra were obtained in information-dependent acquisition mode and were queried against the *Daphnia magna* protein database (http://arthropods.eugenes.org/EvidentialGene/daphnia/daphnia_magna_new/Genes/earlyaccess/) with an in-house Mascot server (version 2.2.1. Matrix Science; [7].

## Microinjection

Following the established protocol for microinjection [5], eggs were obtained from 2–3 week old *D. magna* right after ovulation and were transferred to ice-chilled M4 medium [22] with 80 mM sucrose. An injection marker, 1 mM Lucifer Yellow (Invitrogen, Carlsbad CA, USA), was mixed into the injection cocktail (plasmids, RNAs, and proteins) for each experiment. After injection, the surviving eggs were transferred into each well of 96-well plates which had 100 μL of M4-sucrose medium and were then kept in an incubator at 23˚C.

## CRISPR/Cas-mediated mutagenesis

Guide RNAs (gRNAs) were designed to recognize sequences that code for any of the two RNA Recognition Motifs (RRMs) of the Shep using the ZiFiT software from the website http://zifit.partners.org/ZiFiT/CSquare9Nuclease.aspx. The gRNA sequences were as follows: RRM1 (5´-CGACGACCGGCGGCAGTACC-3´) and RRM2 (5´-ACTTGCCGCCGCACATCACC-3´). To avoid the off-target effects, each gRNA sequence was confirmed to have more than 6 base pair mismatches with the other genes by using the *Daphnia* Genome Database because the DNA region with up to five base pair mismatches with the gRNA is susceptible to editing by the Cas9/gRNA complex [23,24]. These gRNAs were synthesized by the cloning-free method [25] and were transcribed using the MEGAscript T7 kit (Invitrogen, Carlsbad CA, USA). Series of purification procedures then followed: column purification using mini Quick Spin RNA gel columns (Roche Diagnostics, Mannheim, Germany), phenol/chloroform extraction, and ethanol precipitation. Finally, the purified RNAs were dissolved in DNase/RNase-free water and were mixed with Cas9 protein for microinjection into female eggs of *Dsx1* reporter strain as previously described [5].

Somatic mutations of the injected embryos were confirmed by amplification of the target loci from the genomic DNA isolated from each sample. The genomic DNA was extracted by homogenization in 90 μL of 50 mM NaOH with zirconia beads of 1.0 Ø size. Samples were heated at 95 ºC for 10 min, followed by a neutralization and stabilization step by adding 10 μL of 1 M Tris-HCl (pH 7.5) and 2 μL of 5 mM EDTA. Centrifugation followed at 13,000 g for 5 min, before the use of the supernatant as a template for PCR amplification of the target sequences. Using Hot Start Ex Taq Polymerase (Takara Bio, Shiga, Japan), RRM1 and RRM2 regions were

amplified using the primer sets: Forward (5´- AAGGCTACAGCAGCTCGA -3´), Reverse (5´- CCGCGAATGTAGAGGTTG -3´) and Forward (5´- CCCACTAATTTGTACCTGGC -3´), Reverse (5´- CGCATTTCTCTCTGGATTC -3´) respectively, and amplicons were analyzed through native PAGE gel electrophoresis. Moreover, screening for germ-line mutagenesis was done by culturing the offspring of the injected embryos until they produced the next generation. The same genotyping procedure mentioned above was then performed until a positive mutant line was found and established.

## RNAi

Small interference RNAs were designed using the Block-iT RNAi Designer at http://www.invitrogen.com/rnaidesigner.html. The siRNA targeting *Shep* gene sequence is as follows: shep_siRNA (5´-GCCTCCTATCAAGCGTCAA-3´). While for the negative control targeting a random sequence that does not affect the development of the *Daphnia*, this siRNA sequence was used: control_siRNA (5′-GGUUAAGCCGCCUCACAUTT-3′) [26]. Two nucleotides dTdT were added to each 3′ end of the siRNAs. The siRNAs were diluted with the injection marker 1 mM Lucifer Yellow dye (Invitrogen, Carlsbad CA, USA) to have the final concentration of 100 μM and were injected into eggs of the *Dsx1* reporter daphnia strain at 2–3 weeks of age which were destined to be male or female. Samples were then observed at 30 h after injection and collected at 48 h for RNA extraction and cDNA synthesis as previously described [4]. RT-qPCR was then performed to check the expression level changes of the genes of interest (*Shep*, *Dsx1*, and *mCherry*) between the control_siRNA- and shep_siRNA-injected samples.

## Quantitation of the fluorescence

Samples were observed and their photos were taken using Leica DC500 CCD Digital Camera mounted on Leica M165FC fluorescence microscope (Leica Microsystem, Mannheim, Germany). Fluorescence photography was done using GFP and mCherry filters under the following conditions: 1.0 s exposure time, 3.0x gain, 1.0 saturation and 1.0 gamma for GFP and 2.0 s exposure time, 8.0x gain, 1.0 saturation and 1.6 gamma for mCherry. mCherry and GFP fluorescence intensities were calculated using the ImageJ software, following the calculation protocol of a previous study [27]. The total embryo fluorescence of each sample was normalized by the background fluorescence measurement. In addition, Relative Fluorescence Intensity (RFI) was calculated by dividing the total embryo fluorescence of the injected embryos by the uninjected embryos from the same clutch to nullify the differences in auto-fluorescence between embryos from different mothers. The RFIs of the control samples were then compared against the RFI of the treated embryos. At least 5 control and treated embryos were used for quantitation of the fluorescence at 30 h and 48 h post-injection.

## Quantitative RT-PCR

To analyze the temporal changes in *Shep* expression level during embryogenesis, cDNA previously synthesized [28] from male and female daphniids at different time points: 0, 6, 12, 18, 24, 30, 48, and 72 h after ovulation were used. These samples were subjected to RT-qPCR using the cDNA synthesized from the total RNA of daphniids at each stage.

To measure the expression levels of *Shep* and *Dsx1* in RNAi, mutagenesis and overexpression experiments, the cDNAs of each sample were prepared in three replicates for RT-qPCR analysis. mRNA transcripts were measured using Mx3005P Real-Time QPCR System (Agilent Technologies) under the following conditions: 50˚C for 2 min, 95˚C for 10 min, 40 cycles of 95˚C for 15 sec and 60˚C for 1 min and using SYBR Green qPCR SuperMix (Invitrogen, Carlsbad CA, USA) and specific primers designed (S3 Table) to amplify short PCR products (<150

bp). Expressions based on the Ct value during amplification were calculated and normalized by quantitating the expression level of several reference genes: the ribosomal protein *L32*, ribosomal *L8* gene and *Cyclophilin* gene [29]. The geometric mean of the reference genes was calculated for normalization as previously described [30]. The normalized expression levels of the treated samples were then relatively compared to the expression levels of the control to get the final values. Lastly, gel electrophoresis and dissociation curve analysis were performed to confirm the correct amplicon size and the absence of non-specific bands.

## Ectopic expression of intact and deleted Shep binding site

From pCS-EF1a1::Dsx1 exon3 [4], the region of *Dsx1* exon 3 except for the 40 nt sequence which contains the putative binding site of Shep (Shep BS) was removed for the construct of pCS-EF1a1::Shep BS using the following primer set: Forward (5´- GTGTGTGTGTGTGTGTGT TGACGTT -3´) and Reverse (5´- AACACACACACACACACACACCCGGGCATTGTGATTG -3´). This plasmid was then used as a template to delete the potential Shep binding site using the primer set as follows: Forward (5´-GTGTGTGTGTGTTGACGTTTTTCCAATATATAGA TGGAGGC-3´) and Reverse (5´- GCCTCCATCTATATATTGGAAAAACGTCAACACA-CACAC-3´). Embryos injected with each plasmid were compared to embryos injected with pCS-EF1a1::EF1a1 UTR, which only has the *EF1α1* 5´UTR and 3´UTR [4]. These three plasmids (200 ng/µl) were each injected into female eggs of the *Dsx1*-reporter strain. Injected eggs were observed 30 h after injection to observe and calculate for the fluorescence intensity differences.

## RNA synthesis

To prepare the GFP reporter mRNA harboring the *Dsx1α* 5′ UTR, the expression plasmid pEX-A2JI that has the *EF1α1* 3´ UTR and T7 promoter was first synthesized by Eurofins Genomics. Second, the GFP coding sequence of the 4xEcRE-H2B-GFP plasmid [31] was fused with the *EF1α1* 3´ UTR. Third, *Dsx1α* 5′ UTR was amplified with PCR using the pCS-EF1a1:: Dsx1 exon 3 as a template and fused with the *GFP* harboring *EF1α1* 3´ UTR to construct the mRNA template plasmid pEX-Dsx1 5′ UTR::GFP. Fourth, using this plasmid, the potential Shep binding site was removed with the same primer set described above, resulting in the generation of the pEX-Dsx1 5′ UTR mutant::GFP. To overexpress *Shep*, chimeric Shep cDNA harboring the *EF1α1* 5′ UTR and 3´UTR was designed and subcloned downstream to the T7 promoter as described previously [27]. The *Shep* CDS of this plasmid was then replaced with the CDS of *GFP* to serve as control mRNA.

*In vitro* transcription and poly(A) tail addition for all mRNAs were performed using T7 RNA polymerase and Poly(A) Tailing kits, respectively (Ambion, Foster City, CA, USA). The size of synthesized RNAs and length of the attached poly(A) tail were analyzed by denaturing formaldehyde gel electrophoresis and were taken into account for the RNA amounts used for microinjection.

## Luciferase-based *in vitro* translation assay

Luciferase reporter mRNAs were prepared by using pEX-Dsx1 5′ UTR::GFP and pEX-Dsx1 5′ UTR mutant::GFP and replacing its *GFP* CDS with the *Luciferase* gene sequence from pG5luc (Promega Corporation, Madison, WI). 0.1 µM of these mRNAs were then transcribed using the nuclease-treated rabbit reticulocytes lysate (RRL) *in vitro* translation system from Promega. Following the manufacturer's protocol, each reaction contained 70% v/v of RRL, 0.02 mM amino acid mixture, 0.5 U/µL RNase Inhibitor (Nacalai Tesque Inc., Kyoto, Japan), and specific concentrations of the mRNAs based on their molecular size. After denaturing at 65˚C

for 3 min, luciferase reporter mRNAs were added after pre-incubating the RRL *in vitro* translation mixture at 30˚C for 10 min. The assembled reaction was then further incubated at 30˚C for 90 min and stopped by the addition of 60 μM puromycin. Firefly luciferase activity was then observed using LuminoSkan Ascent where 50 μL Bright-Glo Luciferase assay reagent (Promega Corporation, Madison, WI) was added to 3 μL of the translated reaction. The luminescence data were normalized by subtracting the measurements from the *in vitro* translation reaction without any reporter mRNAs. In the different experiments, the *Shep* mRNA, *Shep-FLAG* mRNA, *DAPALR* full RNA (3.6 kb), RNA transcribing the core element of *DAPALR* harboring the Shep binding site (40 nt) and negative controls (*EcR-FLAG* and *GFP* mRNAs) were added together with the reporter mRNAs to test their effect on the translation activity. The full sequence of *DAPALR* and its core element are shown in S6 Fig.

## UV crosslinking and FLAG pulldown assay

3 x FLAG (5´-GACTACAAAGACCACGACGGTGATTACAAAGATCACGACATCGATTA-CAAGGATGACGATGACAAA-3´) was fused to the 3´ end of the *Shep* CDS in pCS-EF1a1:: Shep to make the mRNA template plasmid pCS-EF1a1::Shep-FLAG. Shep-FLAG mRNA was transcribed *in vitro* and poly(A) was added following the same protocol mentioned above. It was then translated using the nuclease-treated rabbit reticulocytes lysate (RRL) *in vitro* translation system from Promega following the same protocol above. The reaction lysate was then divided equally into two tubes wherein 10 μg of the luciferase reporter mRNA with the Shep binding site was added into one tube and the reporter mRNA without the Shep binding site was added into the other. Both treatments were irradiated under ultraviolet (UV) light at 200 mJ/cm$^2$ and were then transferred to a tube containing 50 μL of PVP-treated anti-FLAG M2 Affinity Gel, rotated at 4˚C for 2 h. Washing was done five times using the High-salt wash buffer [50 mM Tris-HCl (pH 7.4), 1 M NaCl, 1 mM EDTA, 1% Igepal CA-630, 0.1% SDS, 0.5% sodium deoxycholate] and the gel was resuspended using PK buffer [100 mM Tris-HcL (pH 7.4), 50 mM NaCl, 10 mM EDTA] with 200 μg of proteinase K for 20 min at 37˚C as previously described [32]. Total RNA extraction [4] was then performed wherein 10 μg of yeast tRNA (Invitrogen, Carlsbad CA, USA) was added as co-precipitant to ensure the collection of a minute amount of RNA, which was followed by cDNA synthesis. RT-qPCR targeting the bait RNAs and the tRNA as a reference gene was conducted using the primer sets enumerated in S3 Table. The geometric mean of the expression levels of the tRNA genes (Met and Phe) was calculated for normalization as previously described [30]. The wildtype *Shep* mRNA and an unrelated mRNA, *EcR-FLAG* were used as negative controls. The normalized expression levels of all samples were then relatively compared to the expression level in the EcR-FLAG pulldown experiment with the RNA that has no Shep binding site to get the final values.

## Supporting information

**S1 Fig. Phylogenetic tree of the RRM domains of the Shep orthologs.** RRMs of Shep orthologs are labeled with red while the Sex-lethal (SXL) RRM is boxed in blue. The percentages of the replicate tree in which the associated taxa clustered together in the bootstrap test (500 replicates) are shown next to the branches. The bar indicates branch length and corresponds to the mean number of the differences (P<0.05) per residue along each branch. Evolutionary distances were computed using the p-distance method.
(TIF)

**S2 Fig. Multiple sequence alignment of the evolutionarily conserved RRM domains of Shep.** Alignment of the RNA Recognition Motifs (RRMs) of the different Shep orthologs from

different organisms. The color is based on the physicochemical property of the amino acid-based on ClustalW. The boxes represent the position of the two RRM regions.
(TIF)

**S3 Fig. Similarity of *DAPALR* overexpression and *Shep* knockdown phenotype. (A)** Ventral view of female and male embryos of *Dsx1* reporter strain injected with control siRNA and *Shep* siRNA and observed at 30 h after injection. mCherry fluorescence allowed visualization of *Dsx1* expression while GFP fluorescence in the nucleus enabled observation of body structures. The bright field images were used to understand the localization pattern of mCherry expression. **(B)** Ventral view of male embryos of *Dsx1* reporter strain injected with control plasmid and *DPALR*-expressing plasmid observed at 30 h after injection.
(TIF)

**S4 Fig. Embryonic stage of generated *Shep* mutant line.** Ventral view of female and male embryos of *Shep* mutant line observed at 30 h after ovulation. mCherry fluorescence allowed visualization of *Dsx1* expression while GFP fluorescence in the nucleus enabled observation of body structures. The bright field images were used to understand the localization pattern of mCherry expression.
(TIF)

**S5 Fig. Diverse phenotype of *Shep* mutants and their genomic mutations. (A)** Ventral and lateral views of the different phenotypes observed after injection of Cas9 and Shep-targeting gRNAs: (from L to R) normal development, delayed development, abnormal development and unhatched egg. Phenotypes of uninjected embryos showing normal development were also shown as control phenotypes at each stage. mCherry fluorescence allowed visualization of *Dsx1* expression while GFP fluorescence in the nucleus enabled observation of body structures. The merged images of mCherry and GFP were used to understand the localization pattern of mCherry expression. Bright field showed photos of embryos taken using visible light. Scale bar = 200 μm. **(B)** PAGE analysis of PCR products by genomic PCR to amplify the region targeted by each RRM-targeting gRNAs. Asterisks show the genomic mutations in RRM1- and RRM2-coding sequences of embryos showing the different phenotypes after Shep mutagenesis.
(TIF)

**S6 Fig. The nucleotide sequence of DAPALR and its core element.** The full sequence of *DAPALR* is shown. Its overlapping region with *Dsx1* 5′ UTR (205 bp) is highlighted in yellow. Colored in red is the 40 nt core element of *DAPALR* harboring the Shep binding site. The blue box indicates the 10 bp of the sequece subjected to deletion of the Shep binding site for the *in vitro* and *in vivo* experiments.
(TIF)

**S1 Table. Full Mass Spectrometry Data.**
(XLSX)

**S2 Table. Summary of mutagenesis experiment.**
(DOCX)

**S3 Table. Primer sequences for RT-qPCR.**
(DOCX)

## Author Contributions

**Conceptualization:** Hajime Watanabe.

**Data curation:** Christelle Alexa Garcia Perez, Shungo Adachi, Quang Dang Nong, Nikko Adhitama, Tadashi Wada.

**Formal analysis:** Nikko Adhitama.

**Funding acquisition:** Yasuhiko Kato, Hajime Watanabe.

**Investigation:** Nikko Adhitama, Tomoaki Matsuura, Toru Natsume, Tadashi Wada.

**Methodology:** Toru Natsume, Yasuhiko Kato.

**Project administration:** Yasuhiko Kato, Hajime Watanabe.

**Supervision:** Nikko Adhitama, Tadashi Wada, Yasuhiko Kato, Hajime Watanabe.

**Validation:** Tomoaki Matsuura, Toru Natsume, Yasuhiko Kato, Hajime Watanabe.

**Writing – original draft:** Christelle Alexa Garcia Perez.

**Writing – review & editing:** Yasuhiko Kato, Hajime Watanabe.

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
