## [Decision Letter · Decision Letter 0]

5 Mar 2021

Dear Dr Watanabe,

Thank you very much for submitting your Research Article entitled 'Sense-overlapping lncRNA as a decoy of translational repressor protein for dimorphic gene expression.' to PLOS Genetics.

The manuscript was fully evaluated at the editorial level and by independent peer reviewers. The reviewers appreciated the attention to an important problem, but raised some substantial concerns about the current manuscript. Based on the reviews, we will not be able to accept this version of the manuscript, but we would be willing to review a much-revised version. We cannot, of course, promise publication at that time.

If you decide to revise the manuscript for further consideration at PLOS Genetics, please aim to resubmit within the next 60 days, unless it will take extra time to address the concerns of the reviewers, in which case we would appreciate an expected resubmission date by email to plosgenetics@plos.org.

[LINK]

We are sorry that we cannot be more positive about your manuscript at this stage. Please do not hesitate to contact us if you have any concerns or questions.

Yours sincerely,

Daniela Delneri

Guest Editor

PLOS Genetics

John Greally

Section Editor: Epigenetics

PLOS Genetics

Editor's comment: The primary issue seem to be the lack of strong biochemical data to infer that shep interacts directly with Dsx or DAPALR. In the revised version, additional experimental evidence that support translational repression is encouraged.

Reviewer's Responses to Questions

**Comments to the Authors:**

Reviewer #1: The manuscript by Perez et al presents a potentially interesting mode of translational regulation via a long non-coding RNA (lncRNA). The authors present data that suggest that the protein shep inhibits Dsx1 alpha isoform translation by the binding to a specific RNA element in the 5’ UTR called the TGE. Interestingly, there is a sense lncRNA called DAPALR, produced from the same region and containing a TGE, which is proposed to sequester shep and would therefore regulate Dsx1 translation. While this represents an interesting role for lncRNA regulation of gene expression, unfortunately the current manuscript is not well presented, the methods are inadequately described and direct evidence of shep binding to the DAPALR and Dsx1 RNA is lacking.

Major comments

1) It would be very useful if Figure 1 had a simple diagram displaying the orientation of the DAPALR lncRNA with the Dsx1 alpha mRNA and an overall layout of the Dsx1 genomic region including the different isoforms.

2) Lines 51-53 – The statement “DAPALR is transcribed from upstream of the transcription start site and overlaps with the 5´ UTR of the Dsx1 alpha isoform, both of which are exclusively expressed in males (Kato et al, 2018, 2011).” needs clarification. Is only the Dsx1 alpha isoform expressed in males and other Dsx1 isoforms exist and are expressed in both males and females? How many Dsx1 isoforms are there and what is their expression in males and females. Does DAPALR only overlap the 5’ UTR of the Dsx1 alpha isoform and not the other isoforms?

3) The description of the RNA pulldown and mass spectrometry methods lack detail (Lines 249-258). The Adachi & Natsume, 2015 reference the authors cite is not open access so it would be difficult for people to see the methods this work is based on. In any case, even with access to this reference key specific details used here are missing. What is the exact 205bp sequence used to make the RNA? How were cell lysates made from female larvae? What were the buffers used for the pulldown, washing and elution? Finally, how was the mass spectrometry carried out? For example, how were pulldown samples prepared for mass spectrometry, what type of mass spectrometry was used, how was the mass spectrometry data analysed and what were the criteria for identifying the three candidate proteins with a higher probability for binding to the overlapping sequence of DAPALR? Also the full mass spectrometry data should be presented as a supplementary file or a link to the data should be given.

4) What is the justification for using female larvae for cell lysates (line 254) when in lines 51-53 it is stated that DAPALR and Dsx1 alpha are exclusively expressed in males? Even though Shep is expressed in both males and females how could you be sure you are not missing a male specific factor that binds to DAPALR and Dsx1?

5) No direct evidence is given for the binding of shep to the TGE like element. Can shep pulldown a DAPALR and Dsx1 RNA with the TGE and when the TGE is mutated is this interaction inhibited? Can shep interact with a TGE RNA in vitro as determined by a technique like EMSA?

6) Additionally, overexpression of the TGE does not directly support the conclusion that the “TGE like motif is essential for Shep and for DAPALR’s function in regulating Dsx1.” How do you know that the TGE is not sequestering another RNA binding protein required for regulating Dsx1 translation?

7) As there is no direct evidence presented for shep binding to a TGE the authors can not call the TGE a “shep binding site”. The experiments presented in Figure 5C and 5D are therefore over interpreted when it is stated that “Shep suppresses the translation by binding to the Dsx1α 5´ UTR”. Can you show that shep does not bind or has reduced binding to an RNA without a TGE?

8) The in vitro luciferase assays presented in Figure 5E require a negative control mRNA in place of the shep mRNA. How do you know that having two mRNAs in the reaction (reporter mRNA and shep mRNA) does not compete for the available ribosomes in the reticulocyte lysate and that is why you have reduced luciferase activity from the reporter mRNA?

9) Mapes et al 2010 showed that the shep C. elegans ortholog, SUP-26, associates with poly(A)-binding protein 1 (PAB-1) in vivo and may repress tra-2 expression by inhibiting the translation-stimulating activity of PAB-1. Does shep interact with the Daphnia PAB-1?

10) A CLIP type technique should be used to determine the range of RNAs that shep binds to and determine whether the shep interacting RNAs are mostly mRNAs or are there other ncRNAs that shep can bind to. It would be interesting to see if all these RNAs contained a TGE. This type of analysis would greatly improve the impact of this manuscript. Is DAPALR the sole example of this mechanism of translational regulation or does this regulation occur more commonly?

Minor comments

The manuscript requires improved English language throughout.

Line 27 – missing word? - determining gene “in” Daphnia magna

Lines 32-34 – sentence is confusing/not clear - “These results suggest that the presence of Shep suppresses the unintentional translation of Dsx1 by setting a threshold and expression of the sense lncRNA, DAPALR cancels the suppression.”

Line 51 – the statement “DAPALR is transcribed from upstream of the transcription start site..” is not clear. Please state what transcription start site you are referring to. I assume it is the Dsx1 alpha transcription start site?

Lines 56-57 – the statement “A series of results showed that Shep functions as a suppressor by binding Dsx1 mRNA 57 and DAPALR sequesters the Shep to activate the dsx1 translation.” is vague. Please specifically state what the “series of results” are.

Lines 106-107 – How can you say DAPALR overexpression causes increased Dsx1 mRNA when you have not specifically measured Dsx1 mRNA by RT-qPCR and have only observed increased mCherry signal in Extended Data Figure 3?

Lines 113-114 – The statement “We used two types of gRNAs targeting each RRM on the Dsx1 reporter strain and obtained a line that has frameshift mutation by 15 nt insertion just before the RRM1 domain” is confusing. This is not a “frameshift” mutation as the correct reading frame is still maintained, it is just an insertion mutation.

Reference needed for the information given in lines 149-151 – Mapes et al 2010

Line 181 – this section and Figure 5E describes an in vitro assay so it is confusing when the authors state that “the Shep suppresses the translation of the injected mRNA (Fig. 5e).” This statement needs to be explained better as I believe the authors are referring back to a previous result.

Line 201 – “is caused by” should be replaced with “causing”

Lines 308-309, line 324 – what were the specific primers used for RT-qPCR of shep, dsx1 and mCherry? Do the dsx1 primers detect all the isoforms?

The Materials and Methods section needs to include the detailed methods of how relative fluorescent intensity differences were obtained and calculated for Figures 2B, 4B and 5D.

Figure 5, panels 5D and E – what do the letters “a” “b” “c” represent above each of the bars in the graphs? This information should be provided in the Figure 5 legend.

Please define somewhere in the methods what the exact sequences of the DAPALR full and DAPALR partial that are used in Figure 5E.

Many of the references are missing their Journal, Volume and/or Pages.

Reviewer #2: In the manuscript Perez et al., the authors identify Shep as an RNA-binding protein of the Daphnia lncRNA DAPALR. Previous work has shown that expression of DAPALR induces the expression of Dsx1, a gene with which it overlaps on the same strand. Both DAPALR and Dsx1 are expressed only in males and are required for male determination. Shep on the other hand is expressed in both males and females. The authors perform a series of experiments using reporter constructs showing that manipulation of Shep levels affect Dsx1 expression but not mRNA levels and conclude that Shep must repress translation of Dsx1 similarly to what has been suggested for the C. elegans Shep homolog Sup-26 translational repression of tra-2. The authors identify a TGE sequence in Dsx1 and DAPALR similar to that identified in tra-2 and perform a series of experiments to suggest that the TGE sequence alone is necessary and sufficient for these effects. Aside from reporter constructs, one in vitro translation experiment is performed in reticulocyte lysate to show that translation of Dsx1 5’ UTR fused to a reporter is sensitive to the presence of Shep and additional DAPALR ncRNA. Overall the results are consistent with the possibility that Shep contributes to translational repression of Dsx1, but no direct biochemical evidence is provided, such as interaction of Shep with translational machinery or changes in Dsx1 mRNA association with translating ribosomes. The results are also consistent with the possibility that DAPALR acts a decoy. The results may be of interest to those studying lncRNA or translation.

Specific points:

1. The authors claim that Shep knockdown results in a pattern that resembles overexpression of DAPALR in females. Do the authors mean males, which is what is shown in Extended Fig. 3?

2. Why is 2x greater Dsx1 transcript observed in females after Shep knockdown in Fig. 2? The explanation of the yolk doesn’t make any sense to me. Are the authors suggesting that Shep regulates Dsx1 expression through a different mechanism specifically in female yolk?

3. Lack of changes of Dsx1 mRNA expression is used throughout the manuscript to argue that the observed effects on reporter expression are due to changes in translation efficiency. The authors should use a second reference transcript other than ribosomal protein L32 to ensure that their results are correctly calibrated.

4. Why do the authors think that the Shep mutant that they generated in Fig. 3 does not increase Dsx1 mRNA levels in females in the same fashion as the Shep knockdown? Is this related to the differences in developmental stage observed? Why aren’t embryos examined for these mutants as in other figures?

5. In Fig. 5, it would be better to use a scrambled sequence for the TGE rather than a full deletion. “Mutated” is a misleading term.

6. In Fig. 5E, the authors should do controls of addition of DAPALR full or partial in the absence of Shep mRNA to ensure that these are Shep-dependent effects.

7. The authors should also consider doing more quantitative experiments to test the decoy model.

Reviewer #3: Comments

In the previous study, the authors' group identified a lncRNA called DAPALR overlapping with the 5'-UTR of the Doublesex1 (Dsx1), which is the male determining gene in Daphnia magna. In the present study, the authors identified the Shep protein as a DAPLAR binding protein and provided a several lines of evidence that Shep functions as a supressor by binding Dsx1 mRNA and DAPLAR sequesters the Shep to activate the dsx1 translation.

Overall, this is a nicely done, and several interesting observations are represented and discussed. However, it remains obscure that the importance of the Shep-DAPALR interaction for governing sexual dimorphic expression of Dsx1. This is because the authors did not mention any about whether functional depletion of Shep by RNAi or CRISPR affects sexual dimorphic trait in the text. Such data was given as supplementary figures but I think these figures should move to the main body of the manuscript and the authors should give some explanations and discussions about the phenotype observed in individuals treated with Shep RNAi and individuals with Shep mutations. Thus, I think major revisions as described below are required prior to the publication.

Major comments for revision

1. Fig. 2A. Did the siRNA mediated knockdown of Shep expression affect sexual development? The author should give some explanations about the phenotype observed in the siShep animals by referring extended figure 3B.

2. Fig. 2C. The expression level of Dsx1 in the siShep females was more than twice as high as that in control females. Was the level comparable to that in normal males? If so, did the siShep females exhibit partial or complete female-to-male sex reversal? I think the authors should describe about such information to help readers to understand the importance of Shep in sexual differentiation. If the siShep females did not show such sex reversal phenotype, then the authors should give some discussions about why knockdwon of Shep expression did not affect sexual development.

3. Fig. 2B~2E. The authors described that RT-qPCR results were normalized with L32 expression levels. However, the values shown in the graph are obviously indicated as relative to the gene expression level in siControl animals. The authors should explain this discrepancy.

4. Lines 106-109. Shep RNAi and DAPALR overexpression in embryos caused increased level in Dsx1 expression in yolk cells. The authors proposed that this may be due to the unique system specialized for energy metabolism in yolk cells. But I don't think this explanation is reasonable for the increased level of Dsx1 expression in yolk cells. The authors should propose more detailed and reasonable mechanisms for how Dsx1 expression was increased by Shep RNAi and DAPALR overexpression in yolk cells.

5. Lines 113-114. In the present study, several lines with mutations in Shep gene were establiShep by CRISPR/Cas9 system. How did the authors rule out the possibility of off-targeting effect. They should provide evidence to prove that the phenotype observed in the Shep mutants were not due to the off-targeting effect.

6. Fig. 3C. The Shep mutation caused increased level of Dsx1 expression in females. Did the Shep mutant females show partial or complete female-to-male sex reversal? The author should clarify this point by referring Extended Data Figure 4 that shows phenotype of the Shep mutants. If the the Shep mutant females did not exhibit any abnormalities in sexual development, then the authors should give some reasons.

7. Fig. 3C, dotted lines. The authors said that the red signal from the guts surrounded by the dotted lines represents the autofluorescence of the food used in daphnia rearing. Then, similar red signal should be observed in males. Why was the red signal only observed in females?

8. Fig. 3D. As shown in Fig. 2C, Shep knockdown caused increased level of Dsx1 expression in female embryos. On the other hand, Shep mutation did not cause such increment in Dsx1 expression level in females. The authors should explain this discrepancy.

9. Lines 128-129. The authors conclude that Shep controls Dsx1 expression at post-transcriptional levels. However, their previous study reported that Dsx1 mRNA level in females was much lower than that in males. If so, the protein level of Dsx1 should be still lower in females than in males even though post-transcriptional suppression by Shep is disrupted by the Shep mutation. The authors should give some explanations about this discrepancy.

10. Extended data Figure 4A. The sex of each individual in the photo should be clearly stated. Also, the authors should make clear whether these individuals show abnormalities in the sexual dimorphic traits. Was the sexual difference in the expression pattern of Dsx1 in these mutants disrupted? The authors should also give some explanations about this to verify whether Shep is indeed responsible for sexual dimorphic expression of Dsx1 protein.

11. Extended data Figure 4B. The images described in this figure are uncomfortable. The authors should load the PCR products on the same gel to precisely compare the size of each band and show a full-picture of the gel to show that each band representing deletion or insertion mutation was specifically observed in the mutant animals. Also the position of the asterisk should be modified to exactly next to the band (in the present version, several asterisks are positioned at where there is no band). The authors should also show the nucleotide sequences of the deletion and insertion mutations as shown in Fig. 3A and clarify whether these mutations cause a frame-shift mutation or not.

Minor comments for revision

1. The entire text and all figures. Regarding the notation of "Shep", the gene symbol should be written as "Shep" in italic notation. The "Shep" notation is only acceptable when it means Shep protein. The same applies to Dsx1.

2. Fig. 2A, 3C, 4A, 5B, and Extended data Figure 5A and 5B. Adding a bright field image will help the reader understand the results.

3. Fig. 3A. Not only show the amino acid sequence encoded by the insertion mutation, but also show the amino acid sequence encoded by the wild-type Shep gene.

4. Line 153. Remove "we overexpressed" because it's redundant in the sentence.

**Have all data underlying the figures and results presented in the manuscript been provided?**

Reviewer #1: **No: **Mass spectrometry methods and data were not provided. How fluorescent intensity differences were obtained and calculated for Figures 2B, 4B and 5D was not presented.

Reviewer #2: Yes

Reviewer #3: Yes

PLOS authors have the option to publish the peer review history of their article (what does this mean?). If published, this will include your full peer review and any attached files.

Reviewer #1: **Yes: **Raymond O'Keefe

Reviewer #2: No

Reviewer #3: No

---

## [Decision Letter · Decision Letter 1]

26 May 2021

Dear Dr Watanabe,

Thank you very much for submitting the revised version of your Research Article entitled 'Sense-overlapping lncRNA as a decoy of translational repressor protein for dimorphic gene expression.' to PLOS Genetics.

The manuscript was fully evaluated at the editorial level and by independent peer reviewers. The reviewers appreciated the improvements, but still raised some substantial concerns about the current manuscript. Based on the reviews, we will not be able to accept this version of the manuscript, but we would be willing to review a revised version. We cannot, of course, promise publication at that time.

Should you decide to revise the manuscript for further consideration here, your revisions should address the specific points made by each reviewer. We will also require a detailed list of your responses to the review comments and a description of the changes you have made in the manuscript. In particular, Reviewer 1 has raised concerns about drawing conclusions in the absence of an experiment that they are recommending, we would encourage you to pay particular attention to this issue.

If you decide to revise the manuscript for further consideration at PLOS Genetics, please aim to resubmit within the next 60 days, unless it will take extra time to address the concerns of the reviewers, in which case we would appreciate an expected resubmission date by email to plosgenetics@plos.org.

[LINK]

We are sorry that we cannot be more positive about your manuscript at this stage. Please do not hesitate to contact us if you have any concerns or questions.

Yours sincerely,

Daniela Delneri

Guest Editor

PLOS Genetics

John Greally

Section Editor: Epigenetics

PLOS Genetics

Reviewer's Responses to Questions

**Comments to the Authors:**

Reviewer #1: The authors have provided an improved manuscript and have addressed almost all my concerns sufficiently with revised text and additional experiments and controls. However, a "direct" interaction between shep and the DAPALR or Dsx1 RNA has still not been proven. The new in vitro pulldown assays (Figure 6B), used to prove a direct interaction, contain rabbit reticulocyte lysate so there is still the possibility that the interaction between shep and the TGE containing RNA is not direct. As such, the authors still cannot convincingly say that there is not another protein from the reticulocyte lysate that may be bridging the interaction between shep and the RNA. There are two experiments the authors could include that would prove a direct interaction between shep and a TGE containing RNA. The first experiment, which was suggested in my previous review, would be to take only purified shep and only a TGE containing RNA and see if they interact in vitro using a technique like EMSA. Alternatively, the authors can repeat the pull down assays they have included here in Figure 6B but include a UV crosslinking step (to capture any direct RNA-protein interactions) and then pull down under conditions that disrupt any indirect interactions (ie with detergent and high salt).

Reviewer #2: The authors addressed the majority of my previous concerns, but the authors still do not show evidence that Shep directly binds endogenous Dsxl transcripts. The new in vitro binding results do support their hypothesis.

I found that their responses to several of Reviewer #3’s comments were not adequately addressed:

A)

5. Lines 113-114. In the present study, several lines with mutations in Shep gene were establiShep by CRISPR/Cas9 system. How did the authors rule out the possibility of offtargeting effect. They should provide evidence to prove that the phenotype observed in the Shep mutants were not due to the off-targeting effect.

Response (Lines 438-446): Only one mutant line was generated in the experiment (Fig 3A, 2B, 2C). The gRNAs targeting the two RRMs were carefully designed and their specificity to Shep could avoid off-target effects because the DNA region with up to five base pair mismatches with the gRNA is susceptible to editing by the Cas9/gRNA complex (Fu et al, 2013; Jiang et al, 2013). This was mentioned in the methods in Lines 438-446.

More than one mutant line should be examined to ensure that off target effects are not an issue.

B)

7. Fig. 3C, dotted lines. The authors said that the red signal from the guts surrounded by the dotted lines represents the autofluorescence of the food used in daphnia rearing. Then, similar red signal should be observed in males. Why was the red signal only observed in females?

Response: The feeding activity and time of taking the photos were different between the female and male samples hence the difference in the red signal caused by the autofluorescence of the chlorella could be observed.

Compared photos should be taken with the same exposures. Alternatively, the relative exposure times could be indicated.

Additional minor comments:

C) Authors claimed multiple times that Dsxl transcripts are exclusively expressed in males but in fact there are transcripts and translated RFP in females.

D) There is no evidence Shep expression peaks at 30hpo, line 111 and there is a mis-callout of figure 1D, line 110.

E) The Shep mutant has additional 5 amino acids, which lead to a 1% increase of the protein size. In figure 3B, it is way more than 1%.

F) Do any Shep antibodies exist that can be used to validate the manipulation of Shep expression?

G) The authors need better discussion of their decoy model – if the lncRNA is expressed at only 10% of Dsxl transcripts and sufficient to function as a decoy, how would Shep be able to inhibit all remaining Dsxl?

H) If the lncRNA indeed works as a decoy, does its expression fluctuate with Shep and/or Dsxl expression in order to properly decoy?

I) Authors should explain briefly why they chose Shep and CUGBP1 while there are a lot of other factors pulled downed as enriched.

J) In figure 4A, why isn’t the RFP expression symmetrical?

K) What are the RNAs used on line 300? Are they Dsxl-related?

L) Why do the authors use images from different angles/stages for almost all figures?

M) Figure 5 still shows the term “mutated”.

Reviewer #3: I think the revised manuscript is now suitable for publication. I am satisfied with the author's responses and plausible explanations.

**Have all data underlying the figures and results presented in the manuscript been provided?**

Reviewer #1: Yes

Reviewer #2: Yes

Reviewer #3: Yes

PLOS authors have the option to publish the peer review history of their article (what does this mean?). If published, this will include your full peer review and any attached files.

Reviewer #1: No

Reviewer #2: No

Reviewer #3: No

Editor's Comments:

Experimental proof of the direct interaction between Shep and Dsx1 RNA should be included in the revised version.

---

## [Decision Letter · Decision Letter 2]

21 Jun 2021

Dear Dr Watanabe,

Thank you very much for submitting your Research Article entitled 'Sense-overlapping lncRNA as a decoy of translational repressor protein for dimorphic gene expression.' to PLOS Genetics.

The reviewers appreciated the revised manuscript that show the direct interaction between Shep and a TGE site.  Reviewer 2 identified some minor concerns that we ask you address in a revised manuscript.

We therefore ask you to modify the manuscript according to the review recommendations. Your revisions should address the specific points made by each reviewer.

[LINK]

Yours sincerely,

Daniela Delneri

Guest Editor

PLOS Genetics

John Greally

Section Editor: Epigenetics

PLOS Genetics

Reviewer's Responses to Questions

**Comments to the Authors:**

Reviewer #1: The addition of a crosslinking step and stringent washing conditions has now convincingly shown a direct interaction between shep and a TGE containing RNA by pulldown. This experiment addresses my last concern.

Reviewer #2: The authors have satisfied the majority of reviewer concerns. Minor comments are indicated below:

E) The authors need to label ladders.

G) The authors need to include something like their response to reviewers in the Discussion.

**Have all data underlying the figures and results presented in the manuscript been provided?**

Reviewer #1: Yes

Reviewer #2: Yes

PLOS authors have the option to publish the peer review history of their article (what does this mean?). If published, this will include your full peer review and any attached files.

Reviewer #1: No

Reviewer #2: No

---

## [Editor Report · Decision Letter 3]

25 Jun 2021

Dear Dr Watanabe,

We are pleased to inform you that your manuscript entitled "Sense-overlapping lncRNA as a decoy of translational repressor protein for dimorphic gene expression." has been editorially accepted for publication in PLOS Genetics. Congratulations!

Yours sincerely,

Daniela Delneri

Guest Editor

PLOS Genetics

John Greally

Section Editor: Epigenetics

PLOS Genetics

Comments from the reviewers (if applicable):

**Data Deposition**

http://datadryad.org/submit?journalID=pgenetics&manu=PGENETICS-D-21-00054R3

**Press Queries**

---

## [Editor Report · Acceptance letter]

21 Jul 2021

PGENETICS-D-21-00054R3 

Sense-overlapping lncRNA as a decoy of translational repressor protein for dimorphic gene expression. 

Dear Dr Watanabe, 

We are pleased to inform you that your manuscript entitled "Sense-overlapping lncRNA as a decoy of translational repressor protein for dimorphic gene expression." has been formally accepted for publication in PLOS Genetics! Your manuscript is now with our production department and you will be notified of the publication date in due course.

With kind regards,

Zsofi Zombor

PLOS Genetics

On behalf of:
